# Ectopic Tumor VCAM-1 Expression in Cancer Metastasis and Therapy Resistance

**DOI:** 10.3390/cells11233922

**Published:** 2022-12-04

**Authors:** Kristen A. VanHeyst, Sung Hee Choi, Daniel T. Kingsley, Alex Y. Huang

**Affiliations:** 1Center for Pediatric Immunotherapy at Rainbow, Angie Fowler AYA Cancer Institute, Division of Pediatric Hematology-Oncology, UH Rainbow Babies and Children’s Hospital, Cleveland, OH 44106, USA; 2Department of Pediatrics, School of Medicine, Case Western Reserve University, Cleveland, OH 44106, USA; 3Case Comprehensive Cancer Center, Cleveland, OH 44106, USA; 4Department of Pathology, School of Medicine, Case Western Reserve University, Cleveland, OH 44106, USA

**Keywords:** vascular cell adhesion molecule-1, VLA4, immunotherapy, pediatric cancer, anti-integrin therapy, osteosarcoma

## Abstract

Vascular Cell Adhesion Molecule-1 (VCAM-1; CD106) is a membrane protein that contributes critical physiologic functional roles in cellular immune response, including leukocyte extravasation in inflamed and infected tissues. Expressed as a cell membrane protein, VCAM-1 can also be cleaved from the cell surface into a soluble form (sVCAM-1). The integrin α4β1 (VLA-4) was identified as the first major ligand for VCAM-1. Ongoing studies suggest that, in addition to mediating physiologic immune functions, VCAM-1/VLA-4 signaling plays an increasingly vital role in the metastatic progression of various tumors. Additionally, elevated concentrations of sVCAM-1 have been found in the peripheral blood of patients with cancer, suggesting the tumor microenvironment (TME) as the source of sVCAM-1. Furthermore, over-expression of VLA-4 was linked to tumor progression in various malignancies when VCAM-1 was also up-regulated. This review explores the functional role of VCAM-1 expression in cancer metastasis and therapy resistance, and the potential for the disruption of VCAM-1/VLA-4 signaling as a novel immunotherapeutic approach in cancer, including osteosarcoma, which disproportionately affects the pediatric, adolescent and young adult population, as an unmet medical need.

## 1. Introduction

Among all childhood and adolescent & young adult (AYA) cancers, osteosarcoma (OS) is the most prevalent aggressive primary malignancy of the bone affecting this age group. Approximately 400–600 patients are diagnosed with OS within this age range each year in the United States of America. Of these patients, 10–20% have metastatic disease at initial presentation. The lung is the most common site of metastasis. Current initial management of OS includes a combination of surgical resection and intensive multi-drug combination chemotherapy. Despite this aggressive therapy, approximately 30–35% of patients with non-metastatic OS at diagnosis suffer disease recurrence [1]. The most common site for disease recurrence is also the lung. Although advances in combination therapy regimens over the last several decades have improved overall survival for patients with non-metastatic OS, there have been no significant improvements in the survival outcome for patients with metastatic disease. Despite aggressive multimodal therapy, the overall outcome for patients with pulmonary metastatic OS remains dismal at 25–30%. For this reason, novel and directed therapy approaches are desperately needed.

## 2. Targeting VCAM-1/VLA-4 Interaction in Osteosarcoma

In an attempt to address this highly unmet clinical need, as illustrated above for children and AYA patients with metastatic pulmonary OS, our research group has composed an IND-approved, single arm, open label, proof of concept clinical trial (NCT03811886) entitled “A Phase I/II Study of Natalizumab as a Single Agent in Children, Adolescents and Young Adults With Recurrent, Refractory or Progressive Pulmonary Metastatic Osteosarcoma, “which is currently enrolling patients age 5–30 at the Angie Fowler AYA Cancer Institute/University Hospitals Rainbow Babies & Children’s Hospital in Cleveland, Ohio. The genesis of this trial stems from several clinical and preclinical observations that formulated the scientific rationale for the clinical trial concept, informed the decision by Biogen Idec’s Medical Advisory Board to provide natalizumab for this first-in-cancer trial, and persuaded the FDA to grant IND approval status for the study. First, scanning the available oncomine database reveals that a majority (41 out of 49) of human OS tissues over-expressed VCAM-1, suggesting the possibility that VCAM-1/VLA-4 signaling may similarly promote tumor survival, metastatic progression and immune evasion in metastatic OS. Second, ongoing preclinical investigations in our laboratory found a high level of surface VCAM-1 expression in the highly pulmonary metastatic subline of the mouse OS model, K7M2, relative to the non-metastatic parental K7 cell line. Third, genetic deletion of VCAM-1 in K7M2, depleting pulmonary MACs expressing VLA-4, or blockade of VCAM-1/VLA-4 via intravenous or intratracheal administration of the anti-α4 antibody dramatically reduces pulmonary OS (pOS) incidence or established metastatic disease burden (manuscript in review) in preclinical mouse models, suggesting that inhibition of VCAM-1/VLA-4 interaction may be of clinical benefit in treating clinical pOS in human patients. Lastly, VCAM-1/VLA-4 as therapeutic clinical targets has also been supported by a myriad of studies in other cancer types. In the following review, we provide a survey of available literature that supports the functional contribution of VCAM-1/VLA-4 interaction in cancer and ongoing attempts to exploit this signaling axis for clinical therapeutic purposes.

## 3. Expression and Biological Function of VCAM-1

VCAM-1/CD106 is a member of the immunoglobulin (Ig) superfamily. Discovered in 1989, VCAM-1 was known as an inducible cell surface glycoprotein primarily on the endothelium, particularly in the setting of inflammatory conditions. VCAM-1 plays an important role in contributing to cellular immune responses [2,3]. In particular, VCAM-1 mediates the rolling and adhesion cascades of circulating leukocytes to and across the endothelium under inflammatory conditions [4,5]. This role is well described in immunologic disorders, such as rheumatoid arthritis and asthma [6]. The integrin α4β1 (VLA-4) was identified in 1990 as the first ligand for VCAM-1 and was required for the firm adhesion of B cells to lymphoid germinal centers [7]. Since then, additional integrins have been described as receptors for VCAM-1, but VLA-4 was the most studied. VLA-4 is found on several cell types, including monocytes and macrophages. In addition to being expressed on the cell membrane, VCAM-1 can also be cleaved from the endothelium or other cell surfaces to a soluble form (sVCAM-1). As such, serum sVCAM-1 levels are increased in patients with immunologic disorders, such as rheumatoid arthritis, collagen disorders and autoimmune disorders, including multiple sclerosis [8]. Interestingly, the serum levels of sVCAM-1 are also increased in patients with various malignancies [9]. Specifically, VCAM-1 has been reported to be over-expressed on many types of cancers, such as breast, gastric, ovarian and melanoma, and purported to play a role in the metastatic progression of these tumors [10,11,12,13,14].

## 4. VCAM-1 Structure and Binding to Integrin

As a type 1 transmembrane protein, VCAM-1 is highly conserved among species. Structurally, VCAM-1 is made up of multiple extracellular Ig-like domains, a single transmembrane domain and a 19 amino acid carboxyl-terminus cytoplasmic tail [3,15]. Various mRNA splice-variants exist in mice and humans (Figure 1). In humans, there are two known splice-variants: one containing seven Ig extracellular domains and the other with six, with the truncated form lacking the fourth extracellular domain [16]. In contrast, murine cells express an isoform containing seven Ig domains, as well as a unique three-Ig domain variant that lacks extracellular Ig domains four through seven. This truncated form is the only variant that expresses a glycophosphatidylinositol (GPI) linker, allowing it to link to GPI for insertion into the plasma membrane. [2,14,16].

As binding partners to protein-based cell adhesion molecules, integrins are surface molecules expressed as heterodimers consisting of alpha and beta subunits. There are 18 known alpha subunits and 8 known beta subunits. These 26 subunits dimerize to form 24 known heterodimer pairs. In brief, integrins are made up of a headpiece motif, which contains the beta-propeller region on the alpha subunit, as well as the β-I and/or an α-I domain. The integrin I domains contain the major binding sites. Besides the headpiece motif, both alpha and beta subunits contain lower structural domains, which are attached to the cell surface, as well as a transmembrane domain and a cytoplasmic tail. For further detail on the integrin structures, consult the article by Luo et al. [17,18,19]. The primary binding partner to VCAM-1 is the α4β1 (CD49D/CD29) integrin, also known as very late antigen-4 (VLA-4) [3,19]. Prior studies have shown that VLA-4 binds to the first and fourth extracellular Ig domains of VCAM-1 [3,20,21], with VLA-4 binding to the fourth Ig domain at a lower affinity as compared to domain 1 binding (Figure 1, red arrows) [20]. The alternative integrin α4β7 competes with VLA-4 for binding to the first domain of VCAM-1 (Figure 1, blue arrows); however, α4β7 has a much lower affinity for VCAM-1 than it does for its main binding partner, the mucosal vascular addressin cell adhesion molecule 1 (MAdCAM-1) [21].

One approach to interrogate functional and physical VCAM-1/VLA-4 binding is to use specific blocking antibodies. A common anti-mouse antibody is the antibody clone PS/2, which targets the α4 (CD49d) subunit of VLA-4. Kamata et al. provided evidence that PS/2 specifically binds to the B2 epitope of α4 in mice. They also showed that binding of VCAM-1 to α4β1 requires the Asp-130 residue on β1, as mutation of this site inhibits binding to VCAM-1 [22]. In humans, the therapeutic monoclonal antibody for the treatment of multiple sclerosis and inflammatory bowel disease, natalizumab, blocks the ability of VCAM-1 to bind to the α4 subunit of VLA-4 in a manner similar to PS/2. Yu et. al. showed that natalizumab binds to the propeller region of α4, resulting in noncompetitive inhibition of VCAM-1 binding, as it blocks binding of certain VCAM-1 conformations. Specifically, natalizumab binding hinders certain orientations of VCAM-1′s second Ig domain; however, natalizumab does not restrict α4 binding to the first Ig domain of VCAM-1 [23].

## 5. VCAM-1 in Tumor Metastatic Potential of Adult and Pediatric Cancers

The contribution of VCAM-1 in tumor metastatic potential was first described in adhesion of melanoma cells to the endothelium [2]. Since then, the over-expression of VCAM-1 has been associated with metastasis in various adult-onset cancers, including breast cancer, gliomas, ovarian cancer and colorectal cancer [6,24,25,26,27,28]. In particular, VCAM-1 expression is up-regulated in the higher grade of clinical gliomas [24], human primary breast tumors [25] and human colorectal cancer tissues [28]. VCAM-1 expression in mesothelium is corelated with shortened overall survival of epithelial ovarian cancer (EOC) patients [26]. High serum levels of VCAM-1 have also been identified in colorectal cancer patients compared with healthy controls [27]. In preclinical breast cancer models, evidence suggests that aberrant expression of VCAM-1 on metastatic breast cancer cells in the lungs attracts VLA-4^+^ macrophages (MACs) into the metastatic tumor niche, whereas the primary, non-metastatic breast cancer cells in the mammary gland express little to no VCAM-1 on their surface. This aberrant VCAM-1 expression in tumors is thought to be driven in part by the NF-kB signaling pathway [29,30,31]. The strong interaction between VCAM-1 and VLA-4 was hypothesized to enable VLA-4^+^ monocytes and MACs to recruit tumor cells into the lung tissue. In this scenario, macrophages have been shown to be critically important in aiding tumor metastasis, starting from the primary tumor site to the metastatic niche [32,33]. The specific proposed interaction between VLA-4 on MACs and VCAM-1 on metastatic tumor cells results in clustering of VCAM-1 molecules on the tumor cell surface, leading to a cascade of intracellular events within the tumor cells, including binding of Ezrin, which induces PI3K, leading to activation of Akt. Activation of Akt subsequently provides survival signals to the metastatic breast cancer cells, thereby allowing them to establish a hospitable tumor niche in the lung tissue [10]. Diminishing VCAM-1 expression of mammary tumors or blocking VCAM-1 and VLA-4 binding by anti-VLA-4 antibody decreases the lung metastatic potential of the breast tumor model [10]. Similar interactions between VCAM-1 in metastatic breast cancer cells and VLA-4+ osteoclast progenitors have also been described in osteolytic lesions of metastatic breast cancer [13]. In thyroid cancer, over-expression of VCAM-1 is associated with a higher incidence of lymph node metastasis [34]. Studies on preclinical models of cervical cancer further implicate that ectopic VCAM-1 over-expression on tumor cell surfaces confers cervical cancer cells the ability to evade active T cell-mediated immunotherapy, supporting a view that ectopic tumor expression of VCAM-1 impairs anti-tumor immunity [12]. Pinho et al. recently observed an up-regulation of VCAM-1 on leukemic stem cells, arguing that VCAM-1 is used by cancer cells to escape immune detection and to promote disease progression in acute myeloid leukemia. Its inhibition or deletion reduces leukemic burden and extends survival [35]. These results support the role of tumor VCAM-1 overexpression in the evasion of immune surveillance. Figure 2 summarizes our current understanding of the VCAM-1/VLA-4 signaling interplay between tumor cells and MACs.

## 6. sVCAM-1 in Tumor Metastatic Potential in Adult-Onset Cancers

Soluble form of VCAM-1 (sVCAM-1) can be formed by cleavage of membrane-bound VCAM-1 from the cell surface by a disintegrin and metalloproteinase (ADMA) domain-containing enzymes including ADAM17, ADAM8 and ADAM9 (Figure 2) [36,37,38,39,40]. sVCAM-1 is detected in the serum of several types of cancer patients and was purported as a staging or prognosis marker in several types of cancer [29,39,40]. Studies have shown that sVCAM-1 levels correlate with metastatic incidence, further suggesting TME as the source of sVCAM-1 [41,42]. This indirect evidence suggests that sVCAM-1/VLA-4 signaling promotes tumor survival and reprograms the local immune landscape and response. VCAM-1 is also aberrantly over-expressed in pancreatic ductal adenocarcinoma [43,44]. A recent study published by Takahashi et al. sought to identify a more reliable biomarker, other than CA 19-9 or imaging in pancreatic cancer, to ascertain when a patient may no longer be responding to current chemotherapy treatment, with the rationale that by the time the chemotherapy is clinically determined to be ineffective, the disease has already progressed. They used a genetically engineered mouse model of pancreatic cancer and human plasma samples from patients with advanced pancreatic cancer to show that sVCAM-1 induced treatment resistance to Gemcitabine by attracting macrophages to the TME; thus, they further argued that sVCAM-1 could be a reliable prognostic biomarker of chemotherapy in these patients [45]. When sVCAM-1 levels were evaluated before the initiation of and four weeks following treatment with Gemcitabine from plasma samples of patients with unresectable pancreatic cancer, those patients with a decrease in the VCAM-1 level showed significantly longer progression-free survival and overall survival than those patients with an increase in their sVCAM-1 level [45]. This finding suggests that sVCAM-1 levels may predict chemoresistance within four weeks of receiving chemotherapy. In support of this, patients with gastric carcinoma exhibit a significantly higher sVCAM-1 concentration in the blood as compared to healthy subjects [11]. There was also a significant elevation of the sVCAM-1 concentration in patients with advanced disease compared to those with lower-stage disease, as well as in patients with lymph node metastasis as compared to those without [11]. Again, these available clinical data suggest that sVCAM-1 levels may be useful as a predictive biomarker for disease burden in cancer patients.

## 7. Role of α4β1 (VLA-4) in Metastatic Potential of Adult and Pediatric Cancers

Similar to VCAM-1, VLA-4 also plays a key role in cellular immune response by mediating leukocyte adhesion and migration. VLA-4 is expressed on multiple cell types, including monocytes, thymocytes and lymphocytes. VLA-4 is also expressed on many tumor cells. Its adhesive function plays a vital role in the metastatic tumor progression of these tumors by allowing transmigration out of the blood circulation. In addition to VLA-4′s contribution to metastasis in this manner, it also plays an important role in tumor angiogenesis via 3 steps [46]. First, stem cells or bone marrow-derived progenitor cells express VLA-4, allowing for progenitor cell homing to VCAM-1 present in the tumor periphery. Second, VLA-4-expressing monocytes or myeloid cells are recruited into the TME after interaction with VCAM-1 and CS-1 fibronectin. Lastly, induction by VLA-4 and VCAM-1 on endothelial cell and pericyte interactions allow for the survival of both cell types during angiogenesis [46].

In neuroblastoma, a common childhood malignancy of immature nerve tissue that most commonly originates in the adrenal glands and exhibits a high metastatic potential, the α4 integrin has been shown to be critical for neural crest cell motility [47,48]. Studies done by Young et al. demonstrated that a high level of α4 is associated with a decreased relapse-free survival, arguing that α4 integrin may serve as a prognostic marker of poor therapeutic response in this disease. While it is unclear regarding the mechanism by which the cytoplasmic domain of α4 is critical for tumor metastasis, the adhesive function of α4 alone was insufficient in promoting metastasis, as evidenced from studies involving truncation mutants [49]. In further support of the important role of the tumor expression of VLA-4 in metastatic potential, over-expression of VLA-4 is linked to tumor progression in melanoma, endometrial cancer, neuroblastoma, pancreatic cancer and leukemia, where VCAM-1 is also up-regulated [43,50,51,52,53,54]. VLA4 is up-regulated in metastatic sarcoma cells [50] and endometrial cancer cells [51], and VLA4 over-expression is correlated with the development of metastases of melanoma [53] and the late stage of neuroblastoma [54], while VCAM is enhanced in human pancreatic cancer [43]. Specifically, VLA-4 has been shown to increase the adhesion of melanoma and sarcoma cells to cytokine-activated endothelium via INCAM-110/VCAM-1 interaction [55]. In studies involving leukemia, Jacamo et al. demonstrated that the interaction between VCAM-1 on bone marrow mesenchymal stromal cells and VLA-4 on leukemic cells played a critical role in NF-κB activation [56]. They argued that targeting NF-κB or VLA-4/VCAM-1 signaling axes could be a clinically relevant mechanism to overcome chemoresistance in bone marrow resident leukemic cells, an observation echoed by other similar studies on B-cell acute lymphoblastic leukemia [57,58,59].

## 8. Disruption of VCAM-1/VLA-4 Interaction as a Therapeutic Target in the Pediatric, Adolescent and Young Adult (AYA) Population

As suggested by our ongoing preclinical functional data and other available clinical studies discussed above, disruption of VCAM-1/VLA-4 using natalizumab may be a feasible therapeutic approach in pediatric and AYA cancers, including OS [60,61]. Currently, there are no available direct inhibitors of VCAM-1. However, clinical investigations utilizing small molecule inhibitors of VLA-4 are ongoing [62,63]. Yet, while VLA-4 inhibitors have been used in the treatment of inflammatory disorders, they are not well studied in the management of cancer. Natalizumab (TYSABRI^™^, Elan Pharmaceuticals and Biogen Idec) is a recombinant humanized IgG4κ monoclonal antibody used as an inhibitor against the α4 integrin, which directly disrupts the interaction between VLA-4 and VCMA-1. It is currently approved by the Food and Drug Administration (FDA) for the management of T-cell-mediated autoimmune disorders (e.g., Crohn’s disease (CD) and multiple sclerosis (MS)), where it has been shown to decrease the rate of clinical relapse in adults. Natalizumab is prescribed in adults at a dose of 300 mg IV infused over one hour every four weeks.

A likely reason that natalizumab has not been explored significantly in patients with adult-onset malignancies is due to the immune suppressive effects of the drug. Natalizumab binds to the α4 subunit of α4β1/VLA-4 and α4β7 integrin, which is expressed on the cell surface of all leukocytes except neutrophils, and disrupts the interaction between VLA-4 and VCAM-1. Therefore, broad administration of natalizumab may potentially limit the adhesion and transmigration of leukocytes. As an FDA-approved drug, natalizumab carries a black box warning, because it can increase the risk of developing progressive multifocal leukoencephalopathy (PML), a progressive and nearly uniformly fatal disease due to reactivation of an opportunistic infection of the brain caused by the JC polyomavirus, which leads to changes in the white matter of the brain. For these reasons, access to natalizumab is restricted through a Risk Evaluation and Mitigation Strategies (REMS) program called the TOUCH prescribing program. Risk factors for the development of PML include duration of natalizumab therapy, particularly if greater than 2 years, and the presence of JC virus antibodies. This risk is increased in patients who have received prior immunosuppression, which patients with cancer frequently have. A study of natalizumab in relapsed/refractory multiple myeloma (NCT00675428) was terminated following a Phase I clinical trial due to low study enrollment. Since then, there have been no additional investigations involving natalizumab for the management of cancer until most recently, as described below.

Besides adult patients with autoimmune disorders, some studies have evaluated the safety, tolerability and effectiveness of natalizumab in pediatric MS patients. In one of the larger studies, 24 pediatric patients whose average age at the time of treatment onset was 14 years old (±2.3 years) were treated with natalizumab at adult dosing of 300 mg IV monthly. Some of the pediatric patients had prior exposure to interferon and/or glatiramer acetate therapies. Only four patients discontinued therapy due to poor tolerance and/or hypersensitivity reaction [64]. A prospective study was performed to examine pediatric patients less than 18 years old from the national MS registry in Kuwait who received natalizumab therapy. Of 32 patients who received natalizumab with a mean number of 34.5 ± 18 infusions, none reported significant adverse events [65]. One retrospective study looked at pediatric patients ages 9.8–17.7 years with MS who received a median number of 17 natalizumab infusions [66]. All patients, with the exception of one who weighed less than 45 kg, received a dose of 300 mg with each infusion. Three out of eight patients who completed 12 months of treatment reported a treatment-related side effect of hypersensitivity reaction, pharyngitis and acute gastroenteritis. Another retrospective study reported pediatric patients with a mean age of 14.6 ± 2.2 years who received a median number of 15 infusions of natalizumab at 300 mg/dose. Eight out of nineteen patients had transient, mild adverse side effects [67]. Based on this information, natalizumab is relatively well tolerated in the pediatric population, with patients experiencing mostly mild, transient adverse side effects overall.

While its use in conjunction with immunosuppressive chemotherapy in cancer patients may not be feasible for this reason, natalizumab may have a role as monotherapy when standard of care therapy has failed. Based on our ongoing new preclinical data suggesting a functional link between VCAM-1 expression on pulmonary metastatic OS and VLA-4 on pulmonary MACs, we have composed a single arm, open label, proof of concept Phase I/II clinical trial (NCT03811886) of natalizumab in patients age 5–30 suffering from unresectable pOS. A correlative objective of the study is evaluating the sVCAM-1 levels in the peripheral blood of enrolled subjects before and during treatment with natalizumab. We await the results of this important clinical trial with great anticipation.

## 9. Conclusions

Over the last three decades, the increasingly important role of VCAM-1 in myriad malignancies has been described, including its interaction with VLA-4 and its role in the metastatic progression of various adult tumors. Accumulating new data implicate the VCAM-1/VLA-4 signaling axis as a potentially attractive therapeutic target in pediatric leukemias and solid tumors [68], which warrants additional preclinical and clinical development. The ongoing Phase I/II clinical trial in unresectable, recurrent and refractory pulmonary OS will help to establish the safety and tolerability of anti-α4 antibody therapy or other therapeutic strategies targeting the VCAM-1/VLA-4 signaling axis in the pediatric and AYA cancer patient population, as well as open the door to additional combination studies in OS and other childhood and adult cancers.

## Figures and Tables

**Figure 1 cells-11-03922-f001:**
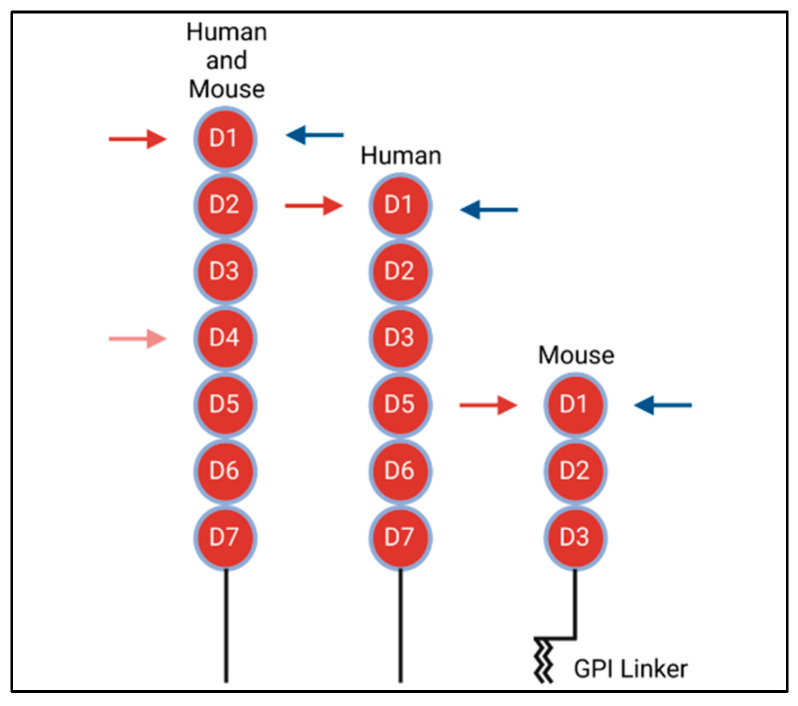
Structures of human and mouse VCAM-1 isoforms. Red circles represent Ig-domains. Red arrows denote VLA4 (α4β1) binding sites on VCAM-1. Blue arrows denote α4β7 binding sites on VCAM-1.

**Figure 2 cells-11-03922-f002:**
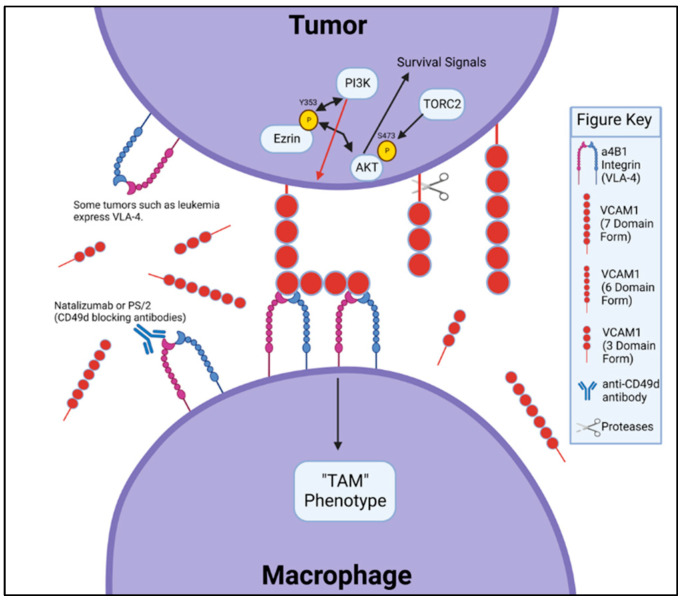
Signaling crosstalk between isoforms of VCAM-1 on tumor surface and VLA4 on macrophages. VLA-4 is also expressed on certain tumors to facilitate metastasis and therapy resistance.

## Data Availability

Not applicable.

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
