# Peer review of "Ectopic Tumor VCAM-1 Expression in Cancer Metastasis and Therapy Resistance"

_cells, 2022, doi:10.3390/cells11233922_

Round 1

Reviewer 1 Report

In this article the authors review the role of VCAM-1, sVCAM-1 and VLA-4 in tumours. But the final aim of the article is to present the clinical trial NCT03811886 that the authors themselves have initiated. 

Frankly, I found the text to be disorganised and it did not lead me to think that the results obtained in this clinical trial could be beneficial for patients (I hope I am wrong, given that patients are already being recruited). I think they need to explain much better what is known so far about VCAM-1 and its interaction with VLA-4 in order to be able to sell their study better.

To give some examples of what I think the authors should improve: 

- It is not clear when they are talking about pre-clinical or clinical studies, nor paediatric or adult tumours. 

- In the section "VCAM-1 in Tumor Metastatic Potential of Adult and Pediatric Cancers" the authors state that VCAM-1 overexpression does not give an advantage over migration but causes TAMs in the lungs to recruit more tumour cells. How are there  tumour-associated macrophages in the lung before there is tumour in that tissue? Why do tumour cells start overexpressing VCAM-1 if primary cells do not express it?

- They also mention several types of cancer in which VCAM-1 and VLA-4 overexpression are associated with tumour progression or worse prognosis, but what type of cells overexpress these markers - tumour cells or inflammatory cells?

- How has sVCAM-1/VLA-4 signalling been shown to promote tumour survival and reprogramme the immune response and landscape? Both mechanisms are different and may explain in different ways the usefulness of targeting therapy against this pathway. 

- It is well understood that the variation in sVCAM levels before starting treatment and after 4 weeks and its relationship to tumour progression is well understood, but how do they demonstrate the attraction of TAMs? with the mouse model? 

- I don't see that the fact that patients with advanced gastric cancer have higher levels of sVCAM supports that sVCAM is a good prognostic biomarker of chemotherapy. 

- About alpha4, is it a good prognostic in neuroblastoma but a poor prognostic in melanoma and sarcoma? it is not clear from the manuscript.

If what the authors are looking for with this review is to demonstrate the importance of the clinical trial they are starting, I think they should to change the order of the text. First they should say how bad osteosarcoma is and the need to look for new treatments. Then say that many of these patients have been found to have VCAM-1 overexpression. Then explain what has been seen in other tumours that have VCAM-1 overexpression and what happens when you reduce VCAM or the VCAM-1 and VLA-4 interaction. Finally, explain that Natalizumab may be useful and that it has been shown to be well tolerated in patients and that is why you are proposing this study. I think that in this way and explaining each of the parts well, the clinical trial may seem interesting.  

Author Response

Reviewer 1:

In this article the authors review the role of VCAM-1, sVCAM-1 and VLA-4 in tumours. But the final aim of the article is to present the clinical trial NCT03811886 that the authors themselves have initiated. Frankly, I found the text to be disorganized and it did not lead me to think that the results obtained in this clinical trial could be beneficial for patients (I hope I am wrong, given that patients are already being recruited). I think they need to explain much better what is known so far about VCAM-1 and its interaction with VLA-4 in order to be able to sell their study better.

We thank the reviewer for this feedback and for providing insightful suggestion to re-order the presentation of various sections in the review, starting with stating the challenges of osteosarcoma, the exciting new clinical trial (NCT03811886), and then extending to a presentation of literature data supporting the targeting VCAM-1/VLA-4 interaction as proposed in the new clinical trial.

To give some examples of what I think the authors should improve:

It is not clear when they are talking about pre-clinical or clinical studies, nor paediatric or adult tumours.

We thank the reviewer and agreed that we have not sufficiently differentiated pre-clinical and clinical studies from pediatric or adult tumors. We have now gone through the revised manuscript to clearly indicate whether studies were pre-clinical vs. clinical studies involving either pediatric or adult tumors.

In the section "VCAM-1 in Tumor Metastatic Potential of Adult and Pediatric Cancers" the authors state that VCAM-1 overexpression does not give an advantage over migration but causes TAMs in the lungs to recruit more tumour cells. How are there tumour-associated macrophages in the lung before there is tumour in that tissue? Why do tumour cells start overexpressing VCAM-1 if primary cells do not express it?

We thank the reviewer for these mechanistic questions. It has been shown in preclinical breast cancer models that migrating breast cancer cells at the metastatic tumor front forms a duplex with a tissue macrophage and co-migrate into blood vessels to distant sites. Once in the metastatic target tissue, the tumor cells contact tissue macrophage to establish metastatic niche. This is now mentioned in Section 5, lines 134-135 (References 32-33) in the revised manuscript. As the new summary Figure 2 depicts, our data current under review for publication also suggest that such interaction polarizes macrophages to the TAM phenotype. This hypothesis was supported by previous published data by Massague and colleagues (Reference 10). Although primary tumors expressed very little VCAM1, the study by Massague and colleagues in preclinical mouse breast cancer model demonstrated that a fraction of the primary tumor cells did upregulate VCAM-1 and this fraction are selected for survival in the new metastatic niche (Figure 2). VCAM-1 is known to be expressed downstream of NF-kB. We now mention this in Section 5, lines 132-133 (References 29-31). It remains unknown whether NF-kB is upregulated in a select small cohort within the primary tumor to over-express VCAM-1 as precursor pro-metastatic cells, or if NK-kB activation and VCAM-1 expression were a secondary phenomenon once the tumors arrived in the metastatic site following stress response which activates NF-kB.  

They also mention several types of cancer in which VCAM-1 and VLA-4 overexpression are associated with tumour progression or worse prognosis, but what type of cells overexpress these markers - tumour cells or inflammatory cells?

We thank the reviewer for pointing the need for clarity on this point. We have now expanded the presentation of the available clinical studies pointing to VCAM-1 expression in various adult cancer cells. The discussions can be found in Section 5, lines 124-129, and Section 7, lines 199-203, with References 6, 24-28, 43, 50-54

How has sVCAM-1/VLA-4 signalling been shown to promote tumour survival and reprogramme the immune response and landscape? Both mechanisms are different and may explain in different ways the usefulness of targeting therapy against this pathway.

In the revised manuscript, we summarize the proposed mechanism of how VCAM-1/VLA-4 interplay between tumor cells and macrophages promote survival and program immune response in the tumor microenvironment. This is now presented as Figure 2.

It is well understood that the variation in sVCAM levels before starting treatment and after 4 weeks and its relationship to tumour progression is well understood, but how do they demonstrate the attraction of TAMs? with the mouse model?

We thank Reviewer 1 for this question. In unpublished preclinical mouse data, we have observed that sVCAM1 levels correlates with metastatic OS disease burden. In vitro, sVCAM was able to polarize macrophages into immune-tolerant phenotypes consisting of increased expression of Arginase-1, CD206, Relma, CD306 (M2-like MAC phenotype) while diminish the expression of iNOS and CD80/CD86, markers associated with M1-like MAC phenotype. These data are currently under review in a manuscript that summarizes our preclinical data, and we chose not present them in this review since they are still under publication consideration.

I don't see that the fact that patients with advanced gastric cancer have higher levels of sVCAM supports that sVCAM is a good prognostic biomarker of chemotherapy.

According to the published paper by Ding et al. (Reference 11), serum sVCAM-1 levels in gastric cancer patients were higher than those in controls and the VCAM-1 positive gastric cancers were more invasive and advanced stage than the VCAM-1 negative stage. Therefore, high serum concentration of sVCAM-1 may serve as a marker of poor gastric cancer burden and response.

About alpha4, is it a good prognostic in neuroblastoma but a poor prognostic in melanoma and sarcoma? it is not clear from the manuscript.

Over-experssion of VLA4 enhances metastasis in multiple cancer types including neuroblastoma, melanoma and sarcoma. We have clarified this point in our discussion of relevant clinical studies (Section 7, Lines 193-205).

If what the authors are looking for with this review is to demonstrate the importance of the clinical trial they are starting, I think they should to change the order of the text. First they should say how bad osteosarcoma is and the need to look for new treatments. Then say that many of these patients have been found to have VCAM-1 overexpression. Then explain what has been seen in other tumours that have VCAM-1 overexpression and what happens when you reduce VCAM or the VCAM-1 and VLA-4 interaction. Finally, explain that Natalizumab may be useful and that it has been shown to be well tolerated in patients and that is why you are proposing this study. I think that in this way and explaining each of the parts well, the clinical trial may seem interesting.

We thank Reviewer 1 for this wonderful suggestion and we wholeheartedly agree. The revised manuscript has now been completely re-organized as suggested by Reviewer 1.

Reviewer 2 Report

In this review article the authors focus on the ectopic expression of VCAM-1 as well as it’s ligand VLA-4 on tumors and the role of this signalling pathway if cancer metastasis. The authors begin by introducing VCAM-1/VLA-4, as well as the structure and binding of the two molecules. They provide evidence of a role for VCAM-1, sVCAM-1 and VLA-4 in metastatic potential. Before concentrating on therapeutic disruption of the signalling pathway, with a focus on pediatric, adolescent and young adult population, more specifically on osteosarcoma. Finally they detail a proof of concept clinical trial that is underway.

I believe that the review article is well written and easy to follow. The authors display thorough knowledge of the topic and the paper is well thought out. The references are appropriate but quite dated (only 15% from the last 5 years), which may just reflect the field. I would encourage the authors to check that there hasn’t been any other more recent advances that they may have missed.

I have some questions for the authors to consider as well as some more specific (minor) details for the authors to address.

·       I think the article would benefit from a summary figure that brings all of the information together to present the overall message of the paper. Ideally a figure that emphasises tumor associated interactions of VCAM-1/VLA-4 and how they promote metastasis (details of specific pathways), compared to the traditional role of VCAM-1/VLA-4 on endothelium and leukocyte rolling.

·       I think it is important for the authors to explain why VCAM-1 is cleaved from the cell membrane to form sVCAM-1. The authors suggest that sVCAM-1 could be used as a prognostic biomarker, and propose to measure this in their clinical trial, so it would be useful for the authors to explain the events that lead to cleavage of sVCAM-1 and why they would expect that to happen in response to natalizumab.

·       Have any of the pediatric patients receiving natalizumab for MS had prior exposure to immunosuppressive therapies and if so did it effect their tolerability? As the authors point out natalizumab is unlikely to be combined with chemotherapy, however it is likely that pediatric patients receiving natalizumab as a monotherapy would have had prior chemotherapy and I wonder how this might influence how well they tolerate the treatment?

·      The authors provide specific details of how natalizumab blocks binding of VCAM-1 to VLA-4 and suggest that only certain orientations of VCAM-1 can bind. This may be a naïve question but does this improve non-specific side effects or help to differentiate normal VCAM-1/VLA-4 interaction from cancer associated interaction?

Minor comments:

·       There are some spelling and grammatical errors, particularly in paragraph 7.

·       Need to define the acronymn pOS (183)

·       Reference #28 Didem et al 2014 does not appear to be about VCAM-1 and does not support the statement made by the authors.

Author Response

Answers to Reviewer 2 comments:

In this review article the authors focus on the ectopic expression of VCAM-1 as well as it’s ligand VLA-4 on tumors and the role of this signalling pathway if cancer metastasis. The authors begin by introducing VCAM-1/VLA-4, as well as the structure and binding of the two molecules. They provide evidence of a role for VCAM-1, sVCAM-1 and VLA-4 in metastatic potential. Before concentrating on therapeutic disruption of the signalling pathway, with a focus on pediatric, adolescent and young adult population, more specifically on osteosarcoma. Finally they detail a proof of concept clinical trial that is underway.

I believe that the review article is well written and easy to follow. The authors display thorough knowledge of the topic and the paper is well thought out. The references are appropriate but quite dated (only 15% from the last 5 years), which may just reflect the field. I would encourage the authors to check that there hasn’t been any other more recent advances that they may have missed.

We thank Reviewer 2 for these comments, and appreciate that Reviewer 2 finds our review article well written and easy to follow. We have gone through literature and added additional references. The total reference count is now 68.

I have some questions for the authors to consider as well as some more specific (minor) details for the authors to address.

I think the article would benefit from a summary figure that brings all of the information together to present the overall message of the paper. Ideally a figure that emphasises tumor associated interactions of VCAM-1/VLA-4 and how they promote metastasis (details of specific pathways), compared to the traditional role of VCAM-1/VLA-4 on endothelium and leukocyte rolling.

We thank Reviewer 2 for this insightful suggestion, and we have now provided a schematic summarizing the interaction between tumor cells and macrophages (Figure 2).

I think it is important for the authors to explain why VCAM-1 is cleaved from the cell membrane to form sVCAM-1. The authors suggest that sVCAM-1 could be used as a prognostic biomarker, and propose to measure this in their clinical trial, so it would be useful for the authors to explain the events that lead to cleavage of sVCAM-1 and why they would expect that to happen in response to natalizumab.

We thank Reviewer 2 for pointing out our lack of mentioning enzymatic mechanisms leading to the production of sVCAM-1 from surface VCAM-1. This is now included in Section 6, lines 156-160. We have also added relevant references (References 36-40) in the revised manuscript.

Have any of the pediatric patients receiving natalizumab for MS had prior exposure to immunosuppressive therapies and if so did it effect their tolerability? As the authors point out natalizumab is unlikely to be combined with chemotherapy, however it is likely that pediatric patients receiving natalizumab as a monotherapy would have had prior chemotherapy and I wonder how this might influence how well they tolerate the treatment?

Some of the pediatric MS patients who have received natalizumab have received Interferon and glatiramer acetate therapies (Reference 64), and have not displayed poor tolerability to the drug. It remains to be seen how prior chemotherapy and the level of immune reconstitution at the time of natalizumab treatment will affect tolerability and side effect profile of natalizumab exposure. A carefully monitored clinical trial such as the one described in the review (NCT03811886) will be instrumental in investigating this aspect of therapy.

The authors provide specific details of how natalizumab blocks binding of VCAM-1 to VLA-4 and suggest that only certain orientations of VCAM-1 can bind. This may be a naïve question but does this improve non-specific side effects or help to differentiate normal VCAM-1/VLA-4 interaction from cancer associated interaction?

We thank Reviewer 2 for this insightful question. This is a question that is yet to be formally answered by experimentation. Our lab is actively investigation this and will report this as soon as we have an insight into this aspect of VCAM-1/VLA-4 biology and therapeutic implication.

Minor comments:

There are some spelling and grammatical errors, particularly in paragraph 7.

We thank Reviewer 2 for this comment. We have now carefully proofread and edited spelling and grammatical errors in the revised manuscript.

Need to define the acronymn pOS (183)

This has now been defined in the revised manuscript, line 63.

Reference #28 Didem et al 2014 does not appear to be about VCAM-1 and does not support the statement made by the authors.

Thank you for catching this mistake. This reference has been removed in the revised manuscript.